# Severe COVID-19 anxiety among adults in the UK: protocol for a cohort study and nested feasibility trial of modified cognitive–behavioural therapy for health anxiety

Mike J Crawford ,[1] Verity C Leeson,[1] Aisling McQuaid,[1] Oluwaseun Samuel,[1] Jacob D King,[1] Martina Di Simplicio,[1] Peter Tyrer,[1] Helen Tyrer,[1] Richard G Watt,[2] Kirsten Barnicot[3]

[1]Division of Psychiatry, Imperial College London, London, UK
[2]Epidemiology and Public Health, University College London, London, UK
[3]School of Health Sciences, City University of London, London, UK

**Correspondence to**
Dr Mike J Crawford;
m.crawford@imperial.ac.uk

## ABSTRACT

**Introduction** Some people are so anxious about COVID-19 that it impairs their functioning. However, little is known about the course of severe COVID-19 anxiety or what can be done to help people who experience it.

**Methods and analysis** Cohort study with a nested feasibility trial with follow-up at 3 and 6 months. We recruited 306 people who were aged 18 and over, lived in the UK and had severe COVID-19 anxiety (indicated by a score of 9 or more on the Coronavirus Anxiety Scale (CAS)). To take part in the nested feasibility trial, participants also had to have a score of 20 or more on the Short Health Anxiety Inventory. We excluded people from the trial if they had had COVID-19 within the previous 4 weeks, if they were currently self-isolating or if they were already receiving psychological treatment.

We publicised the study nationally through adverts, social media and posts on message boards. We also recruited participants via clinicians working in primary and secondary care NHS services in London. All those in the active arm will be offered 5–10 sessions of remotely delivered modified cognitive–behavioural therapy for health anxiety (CBT-HA). We will examine the proportion of participants who remain above threshold on the CAS at 3 and 6 months and factors that influence levels of COVID-19 anxiety over 6 months using mixed effects logistic regression. The key feasibility metrics for the nested trial are the level of uptake of CBT-HA and the rate of follow-up.

**Ethics and dissemination** Approved by Leicester Central Research Ethics Committee (reference: 20/EM/0238). The results of the study will be published in peer-reviewed scientific journals.

**Trial registration number** ISRCTN14973494.

## STRENGTHS AND LIMITATIONS OF THIS STUDY

⇒ The first study to explore the feasibility of using a modified form of cognitive–behavioural therapy for health anxiety for people with severe COVID-19 anxiety.
⇒ Use of remote methods for data collection and delivering the intervention, which enable participation by those who are unable to attend face-to-face meetings.
⇒ The study design means that we will not be able to estimate the prevalence of severe COVID-19 anxiety in the general population.
⇒ Reliance on remote methods for recruitment and follow-up may impact on the follow-up rate.

## INTRODUCTION

The COVID-19 pandemic is having a major impact on mental health throughout the world.[1 2] In the UK, there was a marked increase in the proportion of people experiencing anxiety after the first wave of the pandemic,[3–5] a trend that has also been seen in many other countries.[6] Evidence from longitudinal studies in the UK suggest that a range of factors including social isolation, job insecurity and financial problems were important.[3 7] Qualitative data collected during his period reinforce the importance of these factors and have also highlighted disruption to work and other routines and a sense of lack of control.[8 9]

Fear is an appropriate response to a new infectious disease such as COVID-19 that threatens health, lives and livelihoods. At the start of the pandemic, it was unclear how many people were likely to become infected, what the case fatality rate was and to what extent people who survived the initial illness would go on to experience long-term negative effects. Under these circumstances, fearfulness and anxiety are to be expected. Indeed, research conducted in the early phase of the pandemic found that fear of COVID-19 can be adaptive, resulting in greater adherence to public health measures aimed at limiting the spread of the virus.[10 11] However, as the

pandemic has progressed, it has become clear that some people have become so anxious about COVID-19 that it is impacting on their mental health and social functioning. People with severe COVID-19 anxiety are more likely to report harmful use of alcohol and drugs, hopelessness and suicidal thoughts.[12 13] To date, researchers investigating this topic have used slightly different definitions of 'COVID-19 anxiety'. One of the first teams to investigate the subject, led by Lee *et al* in Virginia (USA), used a narrow definition based on the cognitive, behavioural and physiological components of anxiety.[12 13] More recently teams, such as Albery *et al* in London (UK), have adopted a broader definition that incorporates a wider range of behavioural responses including avoidance, checking and threat monitoring.[14 15] Nonetheless, all those investigating COVID-19 anxiety agree that core symptoms are those found in other anxiety disorders, and it impacts on behaviour and social functioning.

At the start of the pandemic, it was unclear what could be done to help people who appeared to have severe COVID-19 anxiety. In the absence of evidence, governments, professional organisations and mental health services recommended general public mental health measures, such as maintaining contact with family and friends, keeping regular routines and sleeping times, and avoiding drug use and excessive use of alcohol.[16 17] However, in our discussions with patients referred to mental health services who had severe COVID-19 anxiety, we found that some appeared to be suffering from health anxiety. Health anxiety, formerly referred to as hypochondriasis, is a condition in which people become overwhelmed by fears of becoming ill.[18]

Health anxiety was a major contributor to poor mental health during previous pandemics.[19 20] Emerging evidence from public surveys in Europe, Asia and North America have shown it is also contributing to poor mental health during the COVID-19 pandemic.[21–23] People with health anxiety may repeatedly search the internet and other media for information about health conditions, leading to mental distress and impaired functioning.[24 25] People with health anxiety may misinterpret bodily sensations as symptoms of a physical illness and seek reassurance from others.[26] However, symptom monitoring and reassurance seeking may increase the amount of time that people think about being unwell and increase their level of anxiety.[18] During the COVID-19 pandemic, people may be at greater risk of health anxiety because of widespread media coverage, misinformation about COVID-19 and uncertainty about the course of the pandemic.[27] In addition to this, people are being actively encouraged to stay alert for signs of infection and get tested for COVID-19 as soon as possible if even mild signs of infection occur. This approach is an important public health strategy for limiting the transmission of COVID-19,[28] but it may also increase the likelihood of people who are predisposed to being anxious about their health becoming severely anxious to the extent that reduces their quality of life and impacts on their ability to function.[20]

The link between health anxiety and poor mental health seen in previous pandemics is important because health anxiety is treatable. Cognitive–behavioural therapy for health anxiety (CBT-HA) leads to long-term reductions in levels of health anxiety, generalised anxiety and depression.[29 30] In recent years, the intervention has been successfully modified to enable it to be delivered remotely.[31]

As social distancing measures are relaxed and people are able to resume social and occupational roles, it is essential to understand the course and consequences of severe COVID-19 anxiety and examine interventions that could improve people's mental health and functioning. However, at this stage, we do not know the course of COVID-19 anxiety or the contribution that health anxiety plays in maintaining it. While CBT-HA is effective under normal circumstances, we need to find out if it is an appropriate intervention to offer during a global pandemic, when people continue to be asked to be vigilant in spotting the signs of infection in an effort to prevent transmission of the virus.

### Objectives

This study aims to reduce current gaps in knowledge and contribute to the development of a more effective response to health anxiety during pandemics. The study objectives are to:

1. Assess the impact of severe COVID-19 anxiety on people's social functioning and health-related quality of life.
2. Examine the course of severe COVID-19 anxiety over a 6-month period.
3. Identify factors that influence the course of severe COVID-19 anxiety.
4. Test the feasibility of a randomised controlled trial of CBT-HA for people with severe COVID-19 anxiety and health anxiety aimed at improving mental health and social functioning.

## METHODS AND ANALYSIS

The COVID Anxiety Project is a cohort study with a nested feasibility trial. People taking part in the study will be followed up for 6 months. The trial is a researcher-blind, parallel-arm, randomised controlled feasibility trial, which is compliant with Standard Protocol Items: Recommendations for Interventional Trials guidelines.[32]

### Study population and sample

Study participants were recruited from the general public in the UK and from those in contact with primary care and secondary care mental health services in London. We aimed to recruit people who self-identified as being anxious about COVID-19 using a broad range of methods including social media sites (Facebook, Reddit, Instagram and Twitter) and mental health charities (MQ research and Anxiety UK). We publicised the study via 19 primary care practices in north and west London and through colleagues working in secondary care mental health

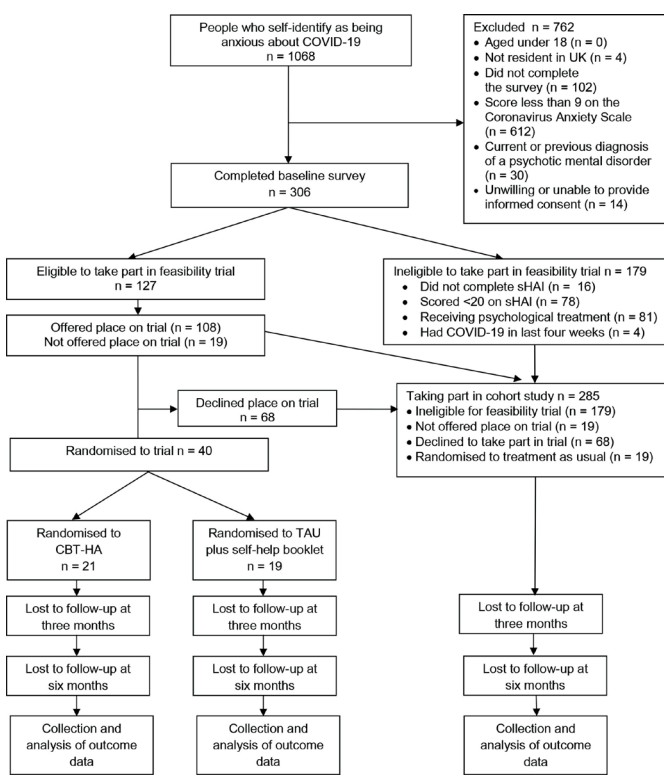

**Figure 1** Study flow diagram. CBT-HA, cognitive–behavioural therapy for health anxiety; sHAI, Short for of Health Anxiety Inventory; TAU, treatment as usual.

services in Central and North West London NHS Foundation Trust, City and East London NHS Foundation Trust and West London NHS Trust.

To be eligible to take part in the cohort study, potential participants had to be aged 18 or over, live in the UK and score 9 or more on the Coronavirus Anxiety Scale (CAS).[12] We used a threshold of 9 or more on the CAS because this identifies those with moderate or severe functional impairment with 90% sensitivity, 85% specificity and a false positive rate of 15%.[12]

We excluded people who had a current or previous diagnosis of a psychotic mental disorder. To take part in the nested feasibility trial, participants also had to have a score of 20 or more on the Short Health Anxiety Inventory (SHAI).[33] We excluded people from the feasibility trial if, at the time of the baseline assessment, they:

▶ Had had COVID-19 in the prior 4 weeks (defined as a positive antigen test or a diagnosis by a clinician).
▶ Were self-isolating on the advice of a doctor or the NHS Test and Trace service.
▶ Were already receiving psychological treatment for any condition.

The cohort study will be made up of all study participants with the exception of those offered CBT-HA as part of the feasibility trial.

### Recruitment and follow-up

Participant flow through the study is presented in figure 1. We posted adverts on social media sites and websites of mental health charities and asked staff working in primary and secondary care NHS services to direct potential participants to a dedicated study website that was hosted by Qualtrics (www.qualtrics.com). The site included a copy of the participant information sheet. Potential participants were asked to sign and date an online consent form, confirm their eligibility to take part in the study and complete the CAS (see online supplemental file 1). The survey was designed so that it could only be completed once from any one IP address.[34] Those who scored 9 or more on the scale and met other eligibility criteria were invited to complete the baseline survey. Those who were not eligible for the study were directed to a webpage which listed sources of mental health support.

We identified potential participants for the feasibility trial from their responses to the baseline survey. We aimed to recruit participants according to the capacity of study therapists to deliver CBT-HA to those in the active arm of the trial. When therapists had capacity to work with more people, we emailed those who recently joined the cohort study and met the additional eligibility criteria a copy of an additional participant information sheet and directed them to an online consent form (see online supplemental file 2). A researcher contacted these participants by telephone to answer any queries they had about the study, confirm that they met study eligibility criteria, support them to complete the online consent form and countersign it. The researcher then contacted the trial manager who randomised the participant. Any participant who was approached to take part in the randomised trial and was ineligible or declined to take part, remained in the cohort study and will be asked to complete subsequent follow-up surveys.

Following attempts by what appeared to be automated survey-takers or 'bots' to take part in the study, we added a CAPTCHA question and a 'trap' question.[34 35] The 'trap' question provided a range of valid and non-valid but plausible options for how the potential participant had heard about the study, with only those providing a valid response being invited to complete the baseline survey.

Between February 2021 and September 2021, we recruited 306 participants to the study. Of these, 127 (41.5%) met inclusion criteria for the feasibility trial. Among 108 participants offered a place on the trial, 40 (37.0%) accepted and were randomised. The total sample for the cohort study is 285 comprising 179 who were ineligible for the trial, 19 who were eligible but not offered a place due to limited capacity of therapists to take on new patients, 68 who were offered a place on the trial but declined it and 19 people who were randomised to treatment as usual.

All participants in the cohort study and the feasibility trial will be sent an automated email to ask them to complete the 3-month and 6-month follow-up survey. Those who have not responded after 1 week will be sent reminders. Trial participants who do not complete follow-up interviews will also be contacted by researchers by email (or telephone if contact number is available) with requests to complete the follow-up surveys.

## Randomisation and blinding

We generated a randomisation list using the independent web-based service 'sealed envelope' (https://www.sealedenvelope.com/simple-randomiser/v1/lists). We stratified according to score on the SHAI (≤24 and ≥25) and the Dependent Personality Questionnaire (DPQ) (≤11 and ≥12) using a ratio of CBT-HA to control treatment of 1:1. We included the DPQ as a stratification variable because previous research has demonstrated that the presence of dependent personality traits may influence the uptake and impact of CBT-HA.[36]

Throughout the study, the randomisation list will be encrypted and stored electronically by the trial manager. At the end of the study, the randomisation list will be unencrypted and placed in the trial master file. The trial manager generated the allocation of each new trial participant by allocating them to the next treatment arm in the predetermined list. The participant was informed of their allocation by telephone or email, and the first therapy session scheduled for those allocated to CBT-HA. Researchers will have very little contact with participants other than to encourage them to complete follow-up surveys. In the unlikely event that researchers speak to a study participant after they have been randomised, they will start by reminding them of the need to make sure they are not aware of whether the participant was offered CBT-HA or control treatment.

## Assessments

The timing and sequence of all assessments are summarised in table 1.

### Screening assessment

To take part in the cohort study, potential participants had to score nine or more on the CAS.[12 13] Developed by Lee in the USA, the scale was designed to provide a brief and reliable measure of the level of COVID-related anxiety that people experience. It comprises five questions on the frequency of anxious thoughts, somatic symptoms and sleep disturbance triggered by reading, thinking or hearing about COVID-19.

### Baseline assessment and covariates

At baseline, we collected self-reported data on demographic factors (age, gender, ethnicity, household composition, occupational status), physical health (current medical conditions) and exposure to COVID-19 (whether they had had COVID-19, whether they had been admitted to hospital with COVID-19). We asked participants whether they live with or care for someone that might get seriously ill if infected with COVID-19 because of their health or age and whether someone in their family or a close friend had been in hospital with COVID-19.

We also asked participants questions about behaviours intended to reduce the risk of exposure to COVID-19. These questions were developed with the help of the members of the Lived Experience Advisory Panel and covered five behaviours which people had used or heard of others using in an effort to avoid catching COVID-19: staying at home, avoiding shops, washing or discarding letters and parcels, increased handwashing and increased washing of clothes. People who lived with school age children were also asked whether they had stopped them attending school because of concerns about COVID-19.

We assessed self-reported functioning using the Work and Social Adjustment Scale[37] and health-related quality of life using the European Quality of Life 5 Dimensions - 3 Levels (EQ-5D-3L).[38] We assessed mental health using the Patient Health Questionnaire-9,[39] Generalised Anxiety Disorder 7-item scale,[40] Obsessive–Compulsive Inventory—Revised,[41] the short form of the Health Anxiety Inventory (SHAI),[33] the Standardised Assessment of Personality—Abbreviated Scale (SAPAS),[42] the DPQ,[43] use of alcohol (Alcohol Use Disorders Identification Test - Concise (AUDIT-C))[44] and a single-question screening test for drug use.[45] Finally, we asked participants to indicate admissions to hospital, contacts with primary and secondary care services using items from the adult service user schedule.[46]

### Follow-up surveys

All those taking part in the cohort study or the feasibility trial are asked to complete follow-up surveys 3 and 6 months after the baseline survey. The content of these surveys was similar to the baseline survey, except that we did not repeat the personality assessments (SAPAS and DPQ), and we added a question at 6 months on whether people had received a COVID-19 vaccination (see table 1).

In recognition of their time and support, all those who completed a baseline interview were sent a £10 gift voucher and a further £20 voucher will be offered to those that complete the 6-month follow-up interview.

### Serious adverse events (SAEs)

Adverse events that are identified in the survey (eg, worsening or newly emerging clinical phenomena) will be added to an adverse event log. Data on SAE including death, hospitalisation and life-threatening events are recorded at baseline, and 3-month and 6-month follow-up. Therapists will also be regularly reminded that they should report all SAEs to the clinical trial team. SAEs will be recorded on a non-CTIMP (Clinical Trials of an Investigational Medicinal Product) SAE form and forwarded within 24 hours to the chief investigator and sponsor if judged to be unexpected and related to the research procedures. The research ethics committee will be notified within 15 days of the chief investigator becoming aware of any such SAE.

### Interventions

All those who agree to take part in the feasibility trial will be sent a self-help booklet developed by staff at Central and North West London NHS Foundation Trust that is derived from the work of Russ Harris on Acceptance and Commitment Therapy,[47] and aims to help people cope with the COVID-19 crisis by building resilience,

**Table 1** Study assessment schedule

| Assessments | Screening | Baseline | 3-month follow-up | 6-month follow-up |
|---|---|---|---|---|
| Coronavirus Anxiety Scale | X | – | X | X |
| Single item psychosis history | X | | | |
| Demographic and clinical data (age, gender, ethnicity, household composition, occupational status, medical history) | – | X | – | – |
| Standardised Assessment of Personality—Abbreviated Scale | – | X | – | – |
| Dependent Personality Questionnaire | – | X | – | X |
| Use of alcohol and drugs (AUDIT-C and the single-question screening test for drug use) | – | X | X | X |
| Generalised Anxiety Disorder 7-item Scale | – | X | X | X |
| Short Health Anxiety Inventory | – | X | X | X |
| Patient Health Questionnaire-9 | – | X | X | X |
| Work and Social Adjustment Scale | – | X | X | X |
| Behaviours intended to reduce the risk of exposure to COVID-19 | – | X | X | X |
| Obsessive–Compulsive Inventory | – | X | X | X |
| Health-related quality of life (EQ-5D-3L) | – | X | X | X |
| Resource use (ADSUS) | – | X | X | X |
| Serious adverse events | – | X | X | X |
| Vaccination status | – | – | – | X |
| Trial participants only: number and length of therapy sessions received | – | – | – | X |

ADSUS, Adult Service Use Schedule; AUDIT-C, Alcohol Use Disorders Identification Test-Concise; EQ-5D-3L, European Quality of Life 5 Dimensions - 3 Levels.

identifying and implementing healthy coping strategies and general advice on health and well-being.[48] All participants will be able to access treatment as usual, which includes access to NHS primary care services and referral on to specialist services if required.

All those in the active arm of the feasibility trial will be offered 5–10 sessions of modified Cognitive Behavioural Therapy for Health Anxiety (CBT-HA) based on a published treatment manual.[49] Therapists will start by taking a detailed history of the person's thoughts and fears about COVID-19 and exploring their beliefs about the impact of COVID-19 and their ability to cope, in order to develop a formulation. Therapists will seek to identify behaviours such as symptom monitoring and reassurance seeking that could be maintaining the person's anxiety. They will then use Socratic dialogue, diary keeping and behavioural experiments to help participants recognise the links between their thoughts and behaviour and explore ways to reduce their anxiety.

All CBT-HA sessions will be delivered by phone or videoconferencing, in accordance with the participant's preference. Sessions will generally last between 30 and 50 min and be offered on a weekly or fortnightly basis. Therapists will consider delaying their final session in order to give people an opportunity to use the techniques and skills they have learnt and reinforce the changes they have made. Sessions will be delivered in keeping with a published treatment manual but modified to meet the difficulties that people in the study are experiencing. Sessions will be supplemented by booklets summarising the causes of health anxiety and how patients can begin to overcome their fears.

Each therapist will record the number and length of sessions they offer participants so that we can calculate the proportion of people who are offered CBT-HA who start and complete treatment and the number of sessions that people receive.

All therapists have a degree in a health-related subject and have previous experience of delivering psychological treatments. All therapists attended a 90 min training session by HT and will receive fortnightly supervision sessions delivered by HT.

The nature of modifications to the original CBT-HA model will be recorded during the study and presented alongside the results of the feasibility trial.

## Sample size

We aimed to recruit sufficient numbers of participants to the cohort study in order to randomise 40 participants to the feasibility trial, which is a typical size for such a trial.[50] A sample of 40 participants would enable us to detect study consent and intervention completion rates of 50%, with 95% CIs of ±11% and ±15%, respectively.

Assuming a rate of follow-up at 6 months of 75%, our cohort of 285 study participants will generate valid data on 213 people. A sample of 196 participants in the cohort study will enable us to estimate a remission rate of 15%±5%, with 95% CI.[51]

## Data analysis

We will start by examining the characteristics of the study sample, including the prevalence of coexisting physical and mental health conditions, levels of disturbance in work and social functioning, quality of life, and actions taken to try to avoid catching COVID-19 using simple descriptive statistics. For the cohort study, we will examine the proportion who remain above threshold for severe COVID-19 anxiety (9 or above on the CAS) and for COVID-19 anxiety (5 or more on the CAS) at 3 and 6 months after completion of the baseline survey. We will examine factors that influence changes in CAS scores over this period using linear regression analysis. The analysis will be performed in two stages. Initially, the separate association between each factor and CAS at each outcome timepoint will be assessed separately. CAS Score at baseline will be included in all these analyses, as the inclusion of this factor will mean that the analyses reflect factors associated with change in CAS from baseline. The second stage of the analysis will examine the joint association between the factors and CAS Score in a multivariable analysis. To restrict the number of variables in this stage of the analysis, only factors showing some association with the outcome ($p<0.2$) from the first stage of the analysis will be considered in the multivariable analyses. A selection procedure (eg, backwards selection) will be considered to identify factors significantly associated with the outcome and to be included in the final model.

We anticipate that there will be little missing data because the platform we designed for data entry requires participants to complete every item of a questionnaire before moving on to the next section. However, where the number of missing items on a given measure is 20% or less, then the missing value for the items will be substituted by the individual's mean score for the remaining items on the scale. If there are more than 20% missing items in the scale the outcome measure will not be calculated for the participant at that time point.

Our criteria for determining the success of the feasibility study, which are based on thresholds used in other recent feasibility and pilot studies,[52 53] are: recruitment of at least 32 participants (80% of the target study sample of 40 participants), uptake of the intervention by at least 60% of participants in the active arm of the trial, and completion of follow-up interviews at 6 months by 75% of study participants. All data will be analysed in SPSS V.2.0 and Stata V.16.1.[54]

## Patient and public involvement

Patients and the public contributed to the development of this proposal. Charlotte Green, who chaired the Patient & Carers Research Interest group at CNWL NHS Foundation Trust, helped develop the proposal and write the lay summary. We presented plans for the study to members of a local patient research interest group via Zoom. Members expressed strong support for the study and eight agreed to join a Lived Experience Advisory Group. Feedback from the group led to us to adding new questions on the impact of COVID-19 anxiety. Members

made suggestions for how to access potential participants, which we adopted. Members of the group also highlighted the importance of tailoring CBT-HA to the needs of individual patients, including making time for them to discuss broader concerns about their mental health and the impact of the pandemic. Members of the Lived Experience Advisory Group will help us prepare a plain English summary of study findings, which we will make available to all study participants and other members of the public through the study website.

## ETHICS AND DISSEMINATION

Plans for the study were approved by Leicester Central Research Ethics Committee (reference: 20/EM/0238). The study will be conducted in accordance with the recommendations for physicians involved in research on human subjects adopted by the 18th World Medical Assembly, Helsinki 1964 and later revisions.

All research activity will be carried out after participants have given consent, and this documented on either an electronic or paper form. For the cohort study, the consent forms will be completed independently by the participant without researcher support but a member of the study team will be available to answer any questions prior to consent being given. Individual consent completed for the cohort study will be linked in Qualtrics to the questionnaires completed by the same participant at each timepoint in the cohort study to ensure that the person that gave consent is completing the assessments.

Separate consent to take part in the clinical trial will be completed by the participant during a telephone call with a member of the research team. The member of the research team that facilitates the completion of the consent process on the telephone will countersign the consent form to document that they were present.

Completed electronic consent forms will be downloaded from Qualtrics, printed and stored in a master file. A copy of their completed consent form(s) will be provided to each participant. Any original paper consent forms will also be stored in the master file. All participants are free to withdraw at any time from the cohort study or clinical trial without giving reasons and without prejudicing further treatment.

The chief investigator will preserve the confidentiality of participants taking part in the study and is registered under the Data Protection Act. Data from the consent forms and questionnaires, including identifiable personal data, will be stored on Qualtrics, an online survey tool. Qualtrics has ISO 27001 certification, meaning that it meets the international standard for managing confidential and sensitive data. All data transfers to Imperial College London use Transport Layer Security encryption. Imperial College will be the data controller and there is a service-level agreement in place between Imperial College and Qualtrics.

Where questionnaires are administered during a telephone interview rather than online, the researcher will type the responses into Qualtrics. Other personal data that is recorded during the study (eg, therapy notes from CBT sessions) will be typed directly onto the secure server of Imperial College London and will be identified using only the participant identifier. Qualtrics allows data erasure and all study data will be deleted from Qualtrics at the end of the study, following extraction for analysis.

Paper documentation will be minimal during this research. An option to complete paper consent form and screening questionnaires was offered but not taken up by any participant.

The results of the study will be published in peer-reviewed scientific journals. We anticipate three key papers examining baseline data, follow-up data and the results of the feasibility trial. We will work with members of our Lived Experienced Advisory Panel to prepare a lay summary of findings from the study which we will post on the study website and will email all study participants the link to this summary.

## DISCUSSION

This study aims to build on the results of previous research that has highlighted the impact of COVID-19 anxiety on mental health during the current pandemic. By collecting prospective data on a sample of people who have high levels of COVID-19 anxiety, we will examine its course over a 6-month period. Having collected detailed information on demographic and clinical factors at baseline we will be able to examine the role these factors play in maintaining COVID-19 anxiety over time. By conducting a nested feasibility trial, we will be able to examine the uptake and acceptability of an intervention for health anxiety, a condition which is strongly linked to anxiety during pandemics and has the potential to benefit people who experience these problems.

This study has a number of strengths and limitations. We believe that it is the first to examine the feasibility of offering people with severe COVID-19 anxiety a psychological intervention for their distress. While many studies have examined the prevalence and aetiology of mental health problems during the COVID-19 crisis, far fewer have started to explore the impact of interventions for mental health problems, and we are not aware of any studies examining the impact of CBT-HA for those who have coexisting health anxiety. By using remote methods for collecting study data and delivering the intervention, we have been able to include people in the study who were so anxious about COVID-19 that they were unlikely to attend face-to-face appointments.

Because we did not use a general population sample for the survey, we will not be able to estimate the prevalence of severe COVID-19 anxiety in the general population. Our reliance on remote means for data collection may have put off some people from taking part in the study. We did make provision for people taking part in the study via telephone interviews, telephone-based therapy and signing paper copies of consent forms. However, people

with limited or no access to the internet may have been put off from taking part.

We are using the CAS as our primary outcome measure.[12] Since the start of the study, other measures have been developed which assess a wider range of behaviours that are associated with COVID-19 anxiety.[14] While we are also asking study participants to describe the frequency of avoidant behaviours associated with COVID-19 anxiety, the psychometric properties of these questions have not been tested.

Our reliance on remote methods for recruitment and following up study participants may have limited our ability to develop a rapport with them and this could, in turn, affect our response rate.[55] We will keep in touch with study participants by email and offer people an honoraria to complete the follow-up surveys in an effort to minimise loss to follow-up.

While the nested clinical trial is large enough to examine the feasibility of a future explanatory trial of CBT-HA for people with severe COVID-19 anxiety it has not been powered to examine the clinical effectiveness of this intervention.

**Acknowledgements** We are grateful to the members of a combined independent oversight committee, which will oversee project governance, data management and review safety data (Dr John Green (chair), Professor Khalida Ismail and Mr Robert Koch). We thank Paul Bassett, freelance statistican, for reviewing the protocol and revising a draft of the Statictical Analysis Plan. We also thank members of the lived experience reference group including Manisha Ahya, Anjie Chhapia, Charlotte Green, Sandra Jayacodi, Niruben Patel and Vikas Sharma.

**Contributors** MC is the chief investigator who conceived the study and led the design of the study and the study protocol and preparation of this manuscript. VL helped design the study protocol, obtained study approvals and contributed to the preparation of this manuscript. AM, OS and JDK recruited study participants contributed to the design of the study and the study protocol and preparation of this manuscript. HT helped to develop the study intervention and contributed to the preparation of this manuscript. MDS, RW, PT and KB contributed to the design of the study, the development of the study protocol and the preparation of this manuscript. JDK and MC developed the statistical analysis plan. All authors read and approved the final manuscript.

**Funding** This research is supported by funding from the NIHR Imperial Biomedical Research Centre (WBPP_PA2902), an National Institute for Health Research Health Technology Assessment Programme (16/157/02) and an NIHR Senior Investigator Application held by MC (NF-SI-0515-10006). The views expressed are those of the authors and not necessarily those of the NIHR or the Department of Health and Social Care. The sponsor of the trial is Imperial College London.

**Competing interests** HT is author of a book on cognitive–behavioural therapy for treating people with health anxiety. Other authors declare that they have no competing interests.

**Patient and public involvement** Patients and/or the public were involved in the design, or conduct, or reporting, or dissemination plans of this research. Refer to the Methods and analysis section for further details.

**Patient consent for publication** Consent obtained directly from patient(s)

**Provenance and peer review** Not commissioned; externally peer reviewed.

**ORCID iD**
Mike J Crawford http://orcid.org/0000-0003-3137-5772

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
