## [Reviewer comments · BMJ Open]

ARTICLE DETAILS

TITLE (PROVISIONAL)	Severe COVID anxiety among adults in the United Kingdom: protocol for a cohort study and nested feasibility trial of modified Cognitive Behaviour Therapy for Health Anxiety.
AUTHORS	Crawford, Mike; Leeson, Verity; McQuaid, Aisling; Samuel, Oluwaseun; King, Jacob D; Di Simplicio, Martina; Tyrer, Peter; Tyrer, Helen; Watt, Richard; Barnicot, Kirsten

VERSION 1 – REVIEW

REVIEWER	Millroth, Philip Uppsala Universitet
REVIEW RETURNED	13-Jan-2022

GENERAL COMMENTS	Thank you for the opportunity to review this protocol, it was overall very clearly written, seemingly in line with standing guidelines on how to conduct studies of this type. The major comment that I have is concerned with the Data Analysis, a section I believe can be improved. I am very well aware that it is a feasibility trial, and that they are not seeking to test a treatment effect (Sims, 2019), but I think the authors, should start to think about these issues already at this stage. But for the larger cohort analyses, these comments are especially important. Major (Data analysis) “We will examine factors that influence changes in scores over this period using linear regression analysis. With a condensed list of covariates we anticipate running mixed-effects linear regression models with maximum likelihood estimation, while fixing time and covariates, and utilising random effects to adjust for subject-specific heterogeneities.” “Initially the influence of covariates on the changes in CAS scores will be examined separately in a series of univariate analyses. Subsequently, the joint association between the covariates and outcome will then be examined using multiple regression.” - I acknowledge the need for data-driven work on this important topic - but in its current stated form, there are some alarm bells ringing here. There are around 10 covariates (Table 1) that the authors intend to look at (at least how I understand it). It sounds like they are going to run a simple univariate regression for each of these covariates (10 covariates = 10 regressions). Thereafter they proceed with using only the significant covariates in the
--

	multiple regression framework. First, regressions entail causal direction; are there previous evidence for any of the covariates to be causally linked with the predicted variable? If so, then could just as well skip this step – but that would leave them with 10 covariates in a multiple regression, probably underpowered even if they focus only on main effects... My proposal would be to take a step back and acknowledge the very exploratory nature of this work and treat it as such. Preferably in a Bayesian framework which in my opinion is much more robust to handling exploratory work. The software JASP provides really simple tools for getting started with Bayesian analyses. 1) Provide a correlation matrix of all relationships. Make sure parametric assumptions are fulfilled, some covariates may not be suitable to include even after transformation. 2) Choose the covariates that fulfill a pre-defined evidence criteria for being related to the predicted variable (e.g., $BF > 3/10/20$, depending on how conservative you want to be). Along with this, include variables that show evidence for possibly being indirectly related (could be moderators or mediators). 3) Run some multi-model inferences over a large model space and report the model averaged results. Also, look at whether a set of models are clearly more probable at explaining the data than other models. Remember to assert that parametric criteria are fulfilled. “Exploratory hypothesis tests to compare study groups will also be conducted. Linear regression will be used to compare mean difference in outcomes between groups, after adjusting for outcome scores at baseline. Mean differences between groups will be reported along with corresponding 95% confidence intervals. It is acknowledged that the study may be unpowered to show statistical significance, and so the feasibility of the trial will not be determined by the results of these tests.” - Study group could be a used as a covariate in the above framework instead of running yet additional analyses. Minor [ ] At times, the concepts of Health Anxiety and Covid Anxiety are used almost interchangeably – but as the authors themselves state, they are not [ ] one could have Covid Anxiety without having general Health Anxiety (but probably not the other way around). In the title they suggest they are looking at Health Anxiety, but the four objectives all state Covid Anxiety. However, the inclusion criteria holds that the participants must have both ($>9 + >20$): a) Please help the readers separate the two concepts (to the extent that they can be separated) better throughout all parts of the protocol.
--	--

	b) Of course too late to do something practical with now, but something that could be discussed is this “double” inclusion criteria (Covid Anxiety AND Health Anxiety). Of course, this is something they can look more at when studying the whole cohort, but then they should elaborate a bit more on that. [ ] I don't see any problems with the criteria for determining the success of the feasibility study (80% of the target study sample, etc), but it would be nice if they offer either a short explanation or reference for why those exact criteria were chosen. Again, thank you for reviewing this important and promising work – looking forward to see how it progresses. Best regards, Philip Millroth Cited references: Sim, J. (2019). Should treatment effects be estimated in pilot and feasibility studies?. Pilot and feasibility studies, 5(1), 1-7.
--	--

REVIEWER	Spada, M London South Bank University, Psychology
REVIEW RETURNED	12-Feb-2022

GENERAL COMMENTS	A study of this kind is of particular value in this moment in time. I commend the authors for wanting to undertake this line of research. I have a series of points for the authors to consider: (1) I am unclear if the sample has already been recruited – it appears to be the case. Please clarify. (2) The link between COVID-19 anxiety and health anxiety is not well operationalised in your paper. Are you arguing that COVID-19 anxiety leads to health anxiety? There is contradictory evidence regarding this assumption. The bridge between COVID-19 anxiety and health anxiety needs to be strengthened to defend your position that adopting a CBT approach for health anxiety is the best course of action. (3) The difficulty with utilising the Coronavirus Anxiety Scale (Lee, 2020) as an outcome is that you may experience a floor effect in terms of scores. Many individuals are not that anxious about the virus anymore (at least from the data coming in from our studies across continents). What they do display, however, is a set of behaviours best encapsulated by the COVID-19 anxiety syndrome (Nikcevic & Spada, 2020; Nikcevic et al., 2021; Albery et al., 2021). Indeed, these behaviours (avoidance, checking, ruminative thinking) are those that would be targeted by the modified CBT approach for health anxiety that you are proposing. Several researchers have argued that COVID-19 anxiety may be an outcome of such behaviours. An epiphenomenon. You could consider assessing the COVID-19 anxiety syndrome in your study. (4) More detail on the adaptation of the CBT protocol from health anxiety needs to be provided. This brings us back to conceptualising what COVID anxiety is. A fear/anxiety response (Lee's concept) or a series of maladaptive behaviours that lock us into the fear/anxiety response (Nikcevic & Spada, 2020). The latter
--

	view is more closely aligned with CBT and metacognitive models of psychopathology. References: Albery, I. P., Spada, M. M. & Nikčević, A. V. (2021). The COVID-19 anxiety syndrome and selective attentional bias towards COVID-19-related stimuli in UK residents during the 2020-2021 pandemic. Clinical Psychology & Psychotherapy, 28(6), 1367-1378. Nikčević, A. V. & Spada, M. M. (2020). The COVID-19 Anxiety Syndrome Scale: Psychometric properties and initial validation. Psychiatry Research, 292, 113322. Nikčević, A. V., Marino, C., Kolubinski, D. C., Leach, D. & Spada, M. M. (2021). Modelling the contribution of the Big Five personality traits, health anxiety, and COVID-19 psychological distress to generalised anxiety and depressive symptoms during the COVID-19 pandemic. Journal of Affective Disorders, 279, 578-584.
REVIEWER	Lunkenheimer , Frederike Ulm University, Department of Clinical Psychology and Psychotherapy
REVIEW RETURNED	14-Feb-2022
GENERAL COMMENTS	Thank you for the interesting study protocol. Comments are attached in the document.

VERSION 1 – AUTHOR RESPONSE

Reviewer one (Millroth)

1) Adopting a Bayesian approach to analysing study data.

> In response to this suggestion we have obtained additional statistical support from Paul Bassett of Stats Consultancy (www.statsconsultancy.co.uk). Dr Bassett helped us revise the analysis plan and we have thanked him for his contribution in the acknowledgments section of the paper. We have highlighted changes to the analysis plan on page 10 and page 11 of the revised manuscript. While we recognise the potential value of Bayesian approaches to analysing health data, we intend to use the more traditional frequentist approach to our analysis. This is also the approach recommended Dr Bassett and one that readers of BMJ Open will be more familiar with.

2) Clarification of the separation between the concepts of Health Anxiety and Covid Anxiety.

> We have redrafted the introduction to the paper to provide more detailed descriptions of COVID, anxiety and health anxiety and explained how the dual aims of the study are to examine the prognosis of severe COVID anxiety and explore the feasibility of using a modified form of CBT for health anxiety for those who experience these problems.

3) I don't see any problems with the criteria for determining the success of the feasibility study

> We have added a justification for thresholds used to judge whether a future explanatory trial would be feasible on page 10 of the manuscript.

Reviewer 2 (Spada)

1) To clarify whether the sample has already been recruited

> The sample has already been recruited. At the start of the third paragraph on page six of the paper, we state that study recruitment ended in September 2021. We have not made any changes to the paper in relation to this feedback.

2) To provide greater justification of the plan to adopt CBT for health anxiety to try to help people with COVID-19 anxiety

> We have added text to the introduction section of the paper explaining that there were no evidence based approaches for helping people with severe COVID anxiety at the start of the pandemic and we selected this approach because of emerging evidence of high levels of health anxiety among people with COVID anxiety (page 3).

(3) The difficulty with utilising the Coronavirus Anxiety Scale (Lee, 2020) as an outcome is that you may experience a floor effect in terms of scores. Many individuals are not that anxious about the virus anymore (at least from the data coming in from our studies across continents). What they do display, however, is a set of behaviours best encapsulated by the COVID-19 anxiety syndrome (Nikcevic & Spada, 2020; Nikcevic et al., 2021; Albery et al., 2021). Indeed, these behaviours (avoidance, checking, ruminative thinking) are those that would be targeted by the modified CBT approach for health anxiety that you are proposing. Several researchers have argued that COVID-19 anxiety may be an outcome of such behaviours. An epiphenomenon. You could consider assessing the COVID-19 anxiety syndrome in your study.

> We are not able to use the measure developed by Professor Spada and colleagues in this study as baseline data have already been collected. At the time of developing the protocol for this study Professor Spada had not published his work on the 'COVID-19 anxiety syndrome'. However, we fully agree that behaviours such as avoidance are important in understanding COVID anxiety and we did include questions about avoidance in the survey (please see page 8). We have also added the helpful references that were suggested by this reviewer.

(4) More detail on the adaptation of the CBT protocol from health anxiety needs to be provided.

> One of the central aims of the feasibility trial is to learn in what ways the CBT-HA model may need to be modified to better support people with severe COVID anxiety. We will be analysing material recorded by the therapists about any modifications they make and we will present these findings alongside the results of the feasibility trial. We have added a sentence to the end of the section on study interventions to explain this (page 10 of the paper).

Reviewer 3

1) Please consider to add a short discussion to your protocol.

> We have added a short discussion section as requested.

2) Reference numbers in the text should be inserted immediately after punctuation (with no word spacing)

> We have corrected any instances where references were not cited in this way.

3) Abstract: - The specification of the planned number of study participants in the abstract is missing.

> We have added this to the abstract

4) An interesting aspect of the introduction is how COVID-19 leads to increased anxiety in the population. Please provide appropriate references for the statements. Your argument will be clearer to the reader if you provide references and evidence.

> We have added new references to the introduction highlighting findings of studies that have examined possible reasons for increased levels of anxiety during the course of the pandemic.

5) Introduction P3 (typographical error)

> We have corrected this typographical error.

6) Objectives: -“This study aims to fill/reduce current’

> We have made the suggested change.

7) Methods: - To add references for the SHAI and single-question screening test for drug use

> These references have been added (page 8).

8) P8: Serious adverse events: Is there monitoring for suicidal ideation? Is there a standardised procedure for suicidal ideation? For ethical and safety reasons, please describe in detail.

> We have added further information about the recording of adverse and serious adverse events to page 9 of the protocol. With the agreement of the sponsor and the Research Ethics Committee suicidal ideation was not considered a serious adverse event. However, suicidal behaviour that resulted in hospital admission or was judged life threatening was considered serious and reported in the way described in the paper.

9) Data analysis: - Please note that due to the small sample size in this feasibility study, the results will have limited statistical power

We have added this as a limitation to a short discussion section at the end of the paper on page 15 of the manuscript.

VERSION 2 – REVIEW

REVIEWER	Millroth, Philip Uppsala Universitet
REVIEW RETURNED	02-Jun-2022

GENERAL COMMENTS	Thank you for well-thought responses to the issues I pointed out, Best of luck, /Philip
---

REVIEWER	Spada, M London South Bank University, Psychology
REVIEW RETURNED	25-May-2022

GENERAL COMMENTS	All issues raised addressed effectively.
--

REVIEWER	Lunkenheimer , Frederike Ulm University, Department of Clinical Psychology and Psychotherapy
REVIEW RETURNED	11-Jul-2022

GENERAL COMMENTS	This study protocol presents a cohort study and nested feasibility study of modified cognitive behavioural therapy for health anxiety (CBT-HA) related to COVID. The study seems timely and empirically relevant as it examines the functionality of health anxiety response in the COVID pandemic. Furthermore, the essential of the feasibility of a randomised controlled trial of CBT-HA for people with severe COVID anxiety and health anxiety may lead to the implementation of a confirmatory trial to test the
---

	effectiveness that could improve mental health and social functioning. I am pleased that the manuscript has gained in quality through the revision based on the reviewer feedback. I wish you continued success with your project.
--	---

VERSION 2 – AUTHOR RESPONSE

Point-by-point response to the Editor's comments and reviewer's comments

> There are NO further reviewer comments

Aside from the clean copy, please also provide a marked copy of your manuscript with 'tracked changes' and upload it under the file designation 'Main Document - marked copy'. This is to show all the changes you have made for your paper.

> Done

Funder Grant# for NIHR Imperial Biomedical Research Centre: You have indicated a funder for your paper. Please ensure to provide an award/grant number for your funder in the main document file and in ScholarOne.

> Added

Please ensure that the funding statement in the ScholarOne system and main document should be the same.

> Changed

Please indicate the corresponding author on your title page along with his/her email address and affiliations.

> Added

Kindly place your table in the main text where the table is first cited.

> Amended accordingly

We have noticed that you have uploaded files under supplemental material. However, we cannot see any citation for the files within the main text. If the files need to be published as a supplementary file, please cite them as 'supplementary file' in the main text and re-upload the files in PDF format.

> Added.

Kindly place your Authors' contribution, Acknowledgements, Funding Statement and Conflict of Interests Statement before your Reference list.

> I do not understand this request - this IS ALREADY the format of the paper.